# Circulation of a Unique *Klebsiella pneumoniae* Clone, ST147 NDM-1/OXA-48, in Two Diverse Hospitals in Calabria (Italy)

**DOI:** 10.3390/antibiotics14020128

**Published:** 2025-01-26

**Authors:** Emanuele Nicitra, Morena Terrana, Dafne Bongiorno, Saveria Dodaro, Francesca Greco, Sonia Greco, Nadia Marascio, Maria Vittoria Mauro, Marta Pantanella, Grete Francesca Privitera, Angela Quirino, Francesca Serapide, Enrico Maria Trecarichi, Valeria Vangeli, Antonio Mastroianni, Giovanni Matera, Alessandro Russo, Stefania Stefani

**Affiliations:** 1Microbiology Section, Department of Biomedical and Biotechnological Science, University of Catania, 95123 Catania, Italymterrana24@gmail.com (M.T.); stefania.stefani@unict.it (S.S.); 2Microbiology and Virology Unit, Annunziata Hospital, 87100 Cosenza, Italy; s.dodaro@aocs.it (S.D.); francesca.greco@aocs.it (F.G.); m.mauro@aocs.it (M.V.M.); 3Infectious & Tropical Diseases Unit, SS Annunziata Hospital, 87100 Cosenza, Italy; grecosonia1976@gmail.com (S.G.); valeriavangeli@gmail.com (V.V.); antoniomastroianni@yahoo.it (A.M.); 4Unit of Clinical Microbiolgy, Department of Health Sciences, “Magna Graecia” University Hospital of Catanzaro, 88100 Catanzaro, Italy; nmarascio@unicz.it (N.M.); pantanella.marta@gmail.com (M.P.); quirino@unicz.it (A.Q.); mmatera@unicz.it (G.M.); 5Department of Clinical and Experimental Medicine, University of Catania, 95123 Catania, Italy; greteprivitera@gmail.com; 6Department of Medical and Surgical Sciences, University “Magna Graecia” of Catanzaro, 88100 Catanzaro, Italy; f.serapide@unicz.it (F.S.); em.trecarichi@unicz.it (E.M.T.); a.russo@unicz.it (A.R.); 7Department of Life Sciences, Health and Health Professions, Link Campus University, 00165 Rome, Italy

**Keywords:** carbapenemase, molecular monitoring, clonal diffusion

## Abstract

**Background/Objectives:** Carbapenem-resistant *Klebsiella pneumoniae* has become endemic in Europe, including in Italy, where its prevalence has risen dramatically, primarily due to epidemic clones harboring metallo-enzymes. This study aims to investigate the dissemination of *K. pneumoniae* strains co-producing OXA-48 and NDM-1 between two hospitals in southern Italy using molecular analyses. **Methods:** A total of 49 *K. pneumoniae* strains, predominantly co-producing OXA-48 and NDM-1, were collected between March and December 2023. Antibiotic susceptibility testing was conducted following EUCAST guidelines. Whole-genome sequencing (Illumina MiSeq) and bioinformatics tools (CARD, CLC Genomics Workbench) were used to identify resistance and virulence genes, capsule loci, and phylogenetic relationships. **Results:** All isolates exhibited multidrug-resistant or extensively drug-resistant profiles, including resistance to ceftazidime/avibactam and meropenem/vaborbactam. Genomic analysis revealed diverse resistance genes such as *bla_OXA-48_*, *bla_NDM-1_*, *bla_CTX-M-15_*, and *bla_SHV_* variants. Virulence genes associated with capsules, fimbriae, and siderophores were widespread. Most strains were classified as ST147 by MLST and contained various plasmids known to carry antimicrobial resistance. Phylogenetic analysis confirmed their clonal relatedness, highlighting the intra-hospital dissemination of high-risk clones. **Conclusions:** High-risk *K. pneumoniae* clones, particularly ST147, pose significant challenges in healthcare settings due to the extensive antimicrobial resistance driven by plasmid-borne resistance genes, including those that co-produce carbapenemases, like *bla_NDM-1_* and *bla_OXA-48_*. Molecular monitoring of these clones is essential for improving targeted infection control strategies, mitigating the spread of multidrug-resistant pathogens, and managing their clinical impact effectively.

## 1. Introduction

Most deaths globally attributed to antimicrobial resistance (AMR) are due to six bacterial species collectively known as ESKAPE pathogens: *Enterococcus faecium*, *Staphylococcus aureus*, *Klebsiella pneumoniae*, *Acinetobacter baumannii*, *Pseudomonas aeruginosa*, and *Enterobacter* spp. [1]. These bacteria are capable of evading the biocidal effects of antimicrobials and are primarily responsible for severe infections in hospitalized or immunocompromised patients. Treatment is particularly challenging due to the limited availability of effective antimicrobial therapies [2]. To combat this global threat, the European Union’s European Medicines Agency approved the clinical use of ceftazidime/avibactam (CZA) in 2016 and meropenem/vaborbactam (MEV) in 2018 [3]. However, due to their continuous exposure to antibiotics acting as a selective pressure, each one of these ESKAPE pathogens have accumulated antibiotic-resistance genes (ARGs), primarily through horizontal gene transfer (HGT) mediated by plasmids or other mobile genetic elements (MGEs). The accumulation of resistance determinants promotes the emergence of multidrug-resistant (MDR) or extensively drug-resistant (XDR) clones that are untreatable with all currently available antibiotics [4]. According to the World Health Organization (WHO), a significant portion of this burden is caused by infections from carbapenem-resistant Enterobacterales (CRE); among these pathogens, carbapenem-resistant *Klebsiella pneumoniae* (CRKP) has emerged as a major concern, becoming widespread and endemic in several European countries [5]. In Italy, CRKP prevalence among invasive isolates surged from 1% in 2009 to 15% in 2010, peaking at 34% in 2016 before slightly dropping to 27% in 2018 [6]. The loss of efficacy of carbapenems, which represent the last-line therapeutic resource for the treatment of infections caused by MDR Gram-negative bacteria, such as *K. pneumoniae*, inevitably leads to a significant increase in morbidity and mortality, particularly for chronically ill patients in intensive care units (ICUs) and long-term care facilities [7]. The emergence of CRKP clones co-harboring multiple genes responsible for carbapenem resistance, such as *bla*_KPC,_
*bla*_OXA_, *bla*_SHV_, and *bla*_NDM-1_, underscores the serious global health threat posed by *K. pneumoniae* [8]. The increasing number of antimicrobial-resistant *K. pneumoniae* infections, especially those caused by extended-spectrum β-lactamase (ESBL) and carbapenemase-producing *K. pneumoniae*, is largely driven by the geographical spread of successful clonal groups (CGs). Of these, CG147 includes three sequence types (STs): ST147, ST273, and ST392. Among these, ST147 has emerged as a high-risk clone responsible for global hospital outbreaks [9]. In this work, 49 OXA-48 and New Delhi metallo-betalactamase 1 (NDM-1)-co-producing *K. pneumoniae* clinical strains collected between March and December 2023 from two hospitals in Calabria, Italy, of which most belonged to ST147, were phenotypically and molecularly characterized to evaluate their spread following their isolation and first-line characterization. The aim of this work is to highlight and to trace the rapid spread of OXA-48- and NDM-1-coproducing *K. pneumoniae*, particularly in hospital settings, and to assess how both phenotypic and molecular characterization could help to trace the acquisition of new resistance mechanisms.

## 2. Results

### 2.1. Sample Characteristics

A total of 49 clinical samples of *K. pneumoniae* co-producing OXA-48 and NDM-1, identified from two different hospitals of Catanzaro and Cosenza were sent to Catania and included in the study. The samples were further characterized as follows: 48/49 (97.9%) were CZA-resistant *K. pneumoniae* (CZA-R Kp) strains, and 1/49 (2.04%) was a CZA-susceptible *K. pneumoniae* (CZA-S Kp) strain (CZ17). We also tested the following antibiotics of interest: meropenem (MEM) and MEV, identifying 47/49 MEM-resistant strains, 2/49 MEM-susceptible strains, 46/49 MEV-resistant strains, and 3/49 MEV-susceptible strains. All isolates were MDR or XDR, and all showed a similar resistance pattern. The resistance profile of the clinical strains to the following antibiotics was characterized as follows: amoxicillin/clavulanate (46/49), piperacillin/tazobactam (49/49), cefepime (49/49), ceftazidime (44/49), imipenem (40/49), aztreonam (34/49), ciprofloxacin (45/49), amikacin (48/49), gentamicin (44/49), and colistin (16/49). However, 33 out of 49 strains were found to be sensitive to colistin. Multi-Locus Sequence Typing (MLST) analysis revealed that all the strains belonged to ST147, except for CS15 (ST846), CS18 (ST34), and CZ17 (ST512). Lipopolysaccharide (LPS) O-antigen was encoded by genes that differentiate between three serotypes: O1/O2v2, O3b, and O3/O3a, which are distinguished by rearrangements detected in the rfb region. Four different capsular polysaccharides (K-locus) were identified: K10, K110, K117, and K107, all of which are linked to the *wzi* gene sequence. The main characteristics, isolation sources, Multi-Locus Sequence Types (MLSTs), and resistome and virulome profiles are shown in Figure 1. Minimum inhibitory concentration (MIC) values of the primary antibiotics tested and the resistance profiles are provided in Appendix A.

### 2.2. Resistome Analysis

Following phenotypic resistance profiling, the corresponding resistance genes were identified through genomic analysis (Figure 1). All the isolated strains exhibited a similar resistance gene profile (Figure 1), highlighting a common core of genes encoding the beta-lactamases present in all the isolated strains. These included *bla*_OXA-48_, *bla*_NDM-1_, *bla*_CTX-M-15_, and *bla*_SHV_. Several oxacillinase OXA variants were detected: *bla*_OXA-48_ in 43 out of 49 strains, *bla*_OXA-9_ in 1 strain, and *bla*_OXA-10_ in 2 strains. All strains harbored *bla*_NDM-1_ genes, except for four isolates. The extended-spectrum beta-lactamase (ESBL) genes *bla*_CTX-M-15_ and *bla*_SHV_ were also identified in these strains. Specifically, *bla*_CTX-M-15_ was in 45 out of 49 strains, while *bla*_SHV_ was found in several variants, with *bla*_SHV-11/67_ and *bla*_SHV-187_ being the most prevalent. Variants such as *bla*_SHV-61_ and *bla*_SHV-123_ were detected in the CS07 and CZ15 strains, respectively. The carbapenemase *bla*_KPC-3_ was identified only in the CZ17 strain, and the metallo-beta-lactamase *bla_VIM-1_* was detected in CS-18. None of the other strains harbored the *bla_KPC_* or *bla_TEM_* genes. All isolates also carried *ampH* (AmpC-related enzyme) and *oqxA-B* (quinolone efflux pump). A variety of additional acquired determinants associated with resistance to aminoglycosides (*aph*, *AAC*, *aad*, *ANT*, *armA*), sulfonamides (*sul1*, *sul2*), trimethoprim (*dfrA12*, *dfrA14*), macrolides (*mphA*, *mphE*, *msrE*), polymyxins (*arnT*, *eptB*), tigecycline (*tetA*), colistin (*LptD*), efflux system pump (*KpnEFG*), and glycopeptides (*BRP(MBL)*) were also variably present. The *fosA6* gene, associated with resistance to fosfomycin, was found in all strains except for CS18, which harbored *fosA5*.

**Figure 1 antibiotics-14-00128-f001:**
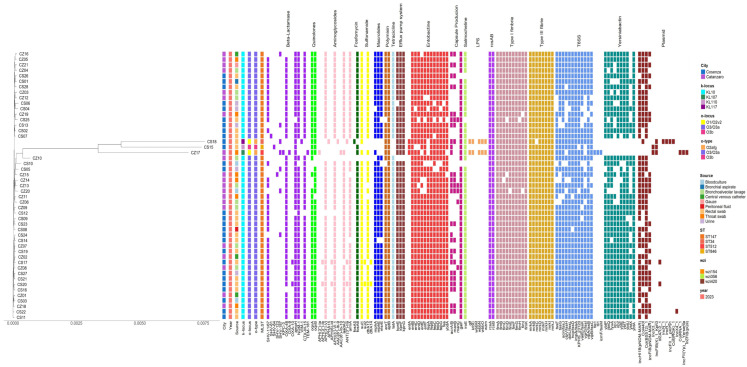
The figure shows the phylogenetic relatedness, as well as the isolation sources, the principal molecular characteristics, and the resistome, virulome, and plasmid profiles. The heatmap was created using the R package ComplexHeatmap (v.2.20.0) [10].

### 2.3. Virulome Analysis

All the isolated strains carried a broad spectrum of virulence factors (Figure 1). Among the enterobactin genes, all the strains possessed the loci *ent*B-C-D-F, *fep*A-B-C-D-G, *fes*, and ybdA; only three strains carried *ent*A. Genes related to capsule production including *gal*F, *acr*A/B, and *man*B, as well as genes involved in the synthesis of the siderophore salmochelin, such as *iro*E, were detected in all the strains. LPS synthesis-related genes, including *glf*, *kfo*C, wbbN, wbbO, and wzm, were present in the CS18 and CZ17 strains. Regarding genes involved in capsular polysaccharide synthesis and the membrane stress response within the *rcs*AB system, *rcs*A and *rcs*B were present in all the strains. A complete set of genes for type I fimbriae (*fim*A-B-C-D-E-F-G-H-I-K) and type III fimbriae (mrkA-B-C-D-F-H-I-J) was found in all isolated strains. Genes for the type VI secretion system (T6SS), including tssF-G and sciN/tssJ, which are part of the core structure, were present in all samples, followed by clpV/tssH, dotU/tssL, hcp/tssD, impA/tssA, KPHS_231, vasE/tssK, vgrG/tssI, vipA/tssB, and vipB/tssC. However, icmF/tssM, tli1, and tle1 were only found in one strain. Yersiniabactin genes were detected as follows: *ybt*P-Q-E-T-U-A-S and *fyu*A in all the strains, *irp*1-2 in 31/49 strains, and *ybt*X in 16/49 strains.

### 2.4. Plasmid Analysis

Multiple plasmids encoding various antimicrobial-resistance genes were identified in all 49 CRKP strains. The plasmid types, identified through PlasmidFinder, were Col440II, IncHI1B (pNDM-MAR), Col (BS512), ColRNAI, IncX3, IncFIB (K)_1_Kpn3, IncA/C2, IncN, IncR, IncFII_1_pk91, Col (IRGK), IncX4, IncFIB (pQil), IncFIB (pNDM-Mar), and IncFII (Yp). Various ARGs were identified from this plasmid that would confer multidrug resistance (MDR), including β-lactamase genes. The IncHI1F (pNDM-MAR) plasmid was found in 20 strains associated with the NMD-1 gene, which confers resistance to β-lactams (Figure 1).

### 2.5. Phylogenetic Tree

Phylogenetic analysis showed that all the ST147 strains, except for CS15-ST846, CS18-ST34, and CZ17-ST512, were closely related (Figure 1). Additionally, the phylogenetic tree indicated that the presence of the same sequence type remained constant and persistent throughout the study period in the two hospital settings. The three ST846-34-512 strains represented sporadic cases isolated from three different patients who had already been colonized by MDR *K. pneumoniae* prior to hospitalization.

## 3. Discussion

In this study, the phenotypic and genotypic features of *K. pneumoniae* strains circulating in two hospital settings in Calabria were evaluated. Most of the carbapenem-resistant clinical strains were isolated and identified from rectal swabs, with all OXA-48/NDM1-producing *K. pneumoniae* strains belonging to ST147, except for three isolates, which did not produce NDM-1 and which belonged to ST846, ST512, and ST34, respectively. Previous studies, such as the CHIMERA project led by Falcone et al. (2022) [11], revealed that among patients with rectal colonization by CR-Kp, the risk of developing bloodstream infections (BSIs) from the same colonizing organism is higher in NDM-Kp rectal carriers compared with KPC-Kp carriers [12]. Therefore, as most of the NDM-Kp strains belonged to ST147, it is likely that carbapenemase types are strongly associated with specific STs [11]. Phylogenetic analysis revealed that all ST147 strains are closely related, as shown in Figure 1. Additionally, the phylogenetic analysis demonstrates that the presence of the same sequence type remained constant and persistent throughout the study period in the two hospital settings. The three ST846-34-512 strains represent sporadic cases isolated from three different patients who were already colonized by MDR *K. pneumoniae* prior to hospitalization.

### 3.1. Resistome

Analysis of the resistome and virulome along with confirmation from the phylogenetic relationships of the strains isolated in the hospital settings of Cosenza and Catanzaro suggests that the high-risk ST147 strains identified likely originated from the expansion of a single clone responsible for causing an outbreak. Almost all the strains carried the *bla_CTX-M-15_* gene, which has become increasingly prevalent over SHV and TEM variants, encompassing a rapidly expanding family that has spread across a wide range of clinically important bacteria and large geographic regions [13]. The presence of *bla_SHV_* genes, including the variants *bla_SHV-11/67_* and *bla_SHV-187_*, further underscores the genetic diversity underlying resistance, with additional ARGs acquired via plasmids exacerbating the MDR and XDR phenotypes and contributing to the resistance to third- and fourth-generation cephalosporins such as ceftazidime and cefepime. Quinolone resistance, detected in all ST147 strains, reflects significant alterations in cellular permeability. These adaptations reduce intracellular antibiotic concentrations, further compromising treatment efficacy [14].

### 3.2. Virulome

Virulome analysis revealed a wide set of virulence determinants in the examined strains (Figure 1). In ST147 strains, the KL10-O3a serotype was predominant. This serotype, previously linked to hospital-acquired infections such as urinary tract infections, pneumonia, and sepsis [9,15], enhances the colonization and infection potential through mechanisms such as siderophore-mediated iron acquisition and fimbriae that facilitate adhesion [16]. Genes encoding the Type VI Secretion System (T6SS) provide additional advantages in competitive environments like the gut and respiratory tract [17].

### 3.3. Plasmids

Wide-host-range plasmids carrying NDM were present in all the CRKP strains, with each hosting at least two plasmids. Epidemic clones like ST147 frequently carried plasmids such as IncHI1B(pNDM-MAR) and Col-like plasmids (e.g., Col4401 or Col [BS512]). These plasmids encode resistance genes, including *bla_CTX-M-15_*, *bla_NDM-1_*, and have also recently been associated with *bla_OXA-48_*, contributing to resistance across multiple antibiotic classes [18]. The mobility of the *bla_NDM-1_* gene, capable of integrating into diverse plasmids (e.g., IncA/C, IncX3, IncHI1), complicates containment and treatment strategies [19]. Notably, the pNDM-MAR plasmid, associated with novel replicons like IncHIB-M and IncFIB-M, co-transports key resistance genes (*bla_NDM-1_*, *bla_CTX-M-15_*, *aac6-1b*), exacerbating the challenge in treating infections caused by these strains [20]. Additionally, the *bla_OXA-48_* gene is predominantly linked to conserved plasmids of the IncL/M type, facilitating efficient conjugation between high-risk *Klebsiella* clones and other species [21].

### 3.4. Final Considerations

The co-production of OXA-48 and NDM-1 in ST147 strains represents a critical threat, given the limited therapeutic options and the frequent emergence of resistance across multiple antibiotic classes. Hospital outbreaks involving ST147 strains co-harboring *bla_OXA-48_* and *bla_NDM-1_* have been reported worldwide, with distinct geographic distributions: *bla_NDM_* is predominant in Southeast Asia and North America, while co-production with *bla_OXA-48_* is more common in Southeast Asia [9,22].The success of ST147 as a high-risk clone lies not only in its resistance determinants but also in their effective dissemination via mobile genetic elements (MGEs) such as plasmids. These MGEs confer a selective advantage by enabling the accumulation of resistance and virulence genes, driving the clonal expansion of epidemic strains. Monitoring plasmids carrying these determinants is crucial, especially in cases of severe outbreaks, for informing containment and treatment strategies. Rodrigues et al. (2022) confirmed the frequent association of *bla_NDM_* and *bla_OXA-48_* with ST147-KL10 isolates, reinforcing the role of these molecular factors in the global success of this clone [9].

## 4. Materials and Methods

### 4.1. Sample Description and Isolation

The study included samples collected from hospitals in Cosenza and Catanzaro (Infectious & Tropical Diseases Unit, Azienda Ospedaliera Cosenza; Infectious & Tropical Diseases Unit, Università “Magna Graecia” Catanzaro) between March and December 2023. A total of 49 *K. pneumoniae* strains, resistant to ceftazidime–avibactam (CZA), were isolated from blood, urine, rectal swabs, bronchial aspirates, peritoneal fluid, and throat swabs. Patients infected by *K. pneumoniae* co-harboring OXA-NMD were included in the study based on tests performed at the time of hospitalization to quickly test for NMD, OXA, and CTX-M positivity. At the Catanzaro hospital, isolation procedures included the following: from blood cultures, the FilmArray Biofire BCID2 Panel (bioMérieux, Marcy-l’Étoile, France); from bronchial aspirates (BAL), the FilmArray Biofire Pneumonia Panel Plus (bioMérieux, Marcy-l’Étoile, France); from throat swabs, chromogenic agar (ESBL, bioMérieux, Marcy-l’Étoile, France) combined with the NG Test/CARBA-5 rapid immunochromatographic test (NG Biotech Laboratories, Guipry, France); and from gauze swabs, the same protocol as for throat swabs. All isolates were identified using MALDI-TOF (Shimadzu, Kyoto, Japan; bioMérieux, Marcy-l’Étoile, France) and Vitek-2 (bioMérieux, Marcy-l’Étoile, France). At the Cosenza hospital, the following methodologies were used by the Microbiology and Virology Unit: for blood cultures, FilmArray Biofire BCID2 Panel (bioMérieux, Marcy-l’Étoile, France) followed by colony confirmation via Vitek MS or Vitek-2 the next day; and for bronchial aspirates (BAL), FilmArray Biofire Pneumonia Panel Plus (bioMérieux, Marcy-l’Étoile, France), with subsequent confirmation as described for rectal surveillance swabs. Cultures were plated on CHROMID CARBA SMART chromogenic media (bioMérieux, Marcy-l’Étoile, France). Colony identification was performed using Vitek MS, and resistance mechanisms were characterized using molecular assays such as eazyplex® SuperBug CRE (AmplexDiagnostics GmbH, Gars-Bahnhof, Germany) or Allplex Entero-DR Assay (Seegene, Seoul, South Korea). All strains of interest, according to the aforementioned criteria, were sent to the Medical Molecular Microbiology and Antibiotic Resistance (MMAR) laboratory at the University of Catania for further molecular characterization. The strains were cultivated on MacConkey agar (Oxoid, CM0007B) and incubated overnight at 37 °C under aerobic conditions before further testing.

### 4.2. Antibiotic Susceptibility Test (AST)

The identification and antimicrobial susceptibility testing of all isolates were preliminarily determined using the aforementioned systems. In addition, minimum inhibitory concentrations (MICs) were determined using the agar diffusion E-test method according to CLSI guidelines (Clinical and Laboratory Standards Institute), using Mueller–Hinton Agar (MHA) (Oxoid, CM0337B) and incubated overnight (o/n) at 37 °C under aerobic conditions. The following antibiotics were tested: ceftazidime (CAZ), meropenem (MEM), amoxicillin/clavulanate (AMC), ceftazidime/avibactam (CZA), meropenem/vaborbactam (MEV), cefepime (FEP), piperacillin/tazobactam (TZP), imipenem (IMI), aztreonam (AZT), meropenem/vaborbactam (MEV), ciprofloxacin (CIP), amikacin (AK), gentamycin (CN), and colistin (COL). Antibiotic breakpoints for the clinical isolates were interpreted according to EUCAST v.13.1, 2023, guidelines for the interpretation of MICs [23].

### 4.3. DNA Extraction

DNA extraction was performed following the instructions provided with the QIAGEN QIAamp^®^ DNA Mini Kit (Ref. 51304; QIAGEN, 40724 Hilden, Germany). DNA was quantified using both the NanoDrop OneC spectrophotometer © 2022 (Thermo Fisher Scientific Inc., Waltham, MA, USA) and the fluorimeter Qubit dsDNA BR Assay Kit (Ref. 32850; Invitrogen, 92008 Carlsbad, CA, USA) to assess the purity and quantity of the initial sample, respectively.

### 4.4. Sequencing

A concentration of 10 ng of each sample was used for NGS sequencing via the Illumina MiSeq platform according to the instructions provided with the QIAseq^®^ FX DNA Library Core Kit (Ref. 1120146; QIAGEN, Hilden, Germany). Libraries were quantified and their quality evaluated using both the fluorometric Qubit dsDNA HS Assay Kit (Ref. Q32851; Invitrogen, Carlsbad, CA 92008, USA) and the Agilent^®^ High Sensitivity DNA Kit (Ref. 5067-4626). Denaturation and dilution of the libraries were performed following the *Denature and Dilute Libraries Guide* protocol provided by Illumina^®^, using 8.5 pM as the loading concentration. Finally, sequencing was performed using the MiSeq Reagent Kits v3 (Ref. 15043895; Illumina, Inc., 92122, San Diego, CA, USA). The Sample Sheet was created using Local Run Manager v3 software and following the instructions in the *Local Run Manager v3 Software Guide* provided by Illumina [24].

### 4.5. Data Analysis

Data were analyzed using the QIAGEN CLC Genomics Workbench software (v24.0.1), following the *User Manual for the CLC Microbial Genomics Module v22.0*, released on 4 January 2022 (QIAGEN, Aarhus, 8000 Denmark). This software uses the CARD (Comprehensive Antibiotic Resistance Database) database to assign resistance, virulence (https://www.mgc.ac.cn/VFs/, accessed on 20 January 2025), and MLST genes (https://card.mcmaster.ca, accessed on 20 January 2025), as described previously [25]. Analysis with the QIAGEN CLC Genomics Workbench software (v24.0.1), following the *User Manual for CLC Microbial Genomics Module v22.0*, released on 4 January 2022 (QIAGEN, Aarhus, 8000 Denmark), completed the protocol. The software defined the resistance, virulence, and MLST genes. The bioinformatic analysis used TrimGalore (v0.5.0) [26,27], to remove the adapter sequence. Furthermore, Unicycler (v0.4.8) and the Illumina-only assembly modality allowed for de novo bacterial sequence assembly. Additionally, the Kleborate (v2.2.0) and Kaptive (v1.3.0) [28,29] commands contributed to the identification of virulence factors, resistance genes, and capsule loci in *Klebsiella pneumoniae*. Plasmid analysis was performed employing the Bactopia software (v1.4.0) [30] using its “bactopia tools” plasmid finder. In silico detection and typing of plasmids was performed using PlasmidFinder and plasmid multi-locus sequence typing [31]. The phylogenetic tree was constructed using mafft (v7.525) [18,19] with 100 iterations, followed by fasttree(v.2.1.11) [20]. The illustration of the tree has been created with the R packages: ape (v5.5), ggtree (v3.2.1), RColorBrewer (v1.1.3), ggplot2 (v3.3.5) and ggnnewscale (v0.5.0.9000) [32,33]. The heatmap was created using the R package ComplexHeatmap (v.2.20.0) [10].

## 5. Conclusions

The burden of *K. pneumoniae* is largely associated with its adaptation to hospital settings and the dissemination of high-risk epidemic clones that harbor numerous resistance determinants, resulting in infections with limited treatment options. Factors such as the pathogen’s ability to colonize the intestinal tract and host tissues and the extensive and indiscriminate use of antibiotics act as selective forces on species such as *K. pneumoniae,* which exhibits a remarkable capacity to accumulate diverse ARGs, often located on epidemic plasmids that promote the dissemination of successful MDR clones. The plasmid analysis highlights the impressive diversity of replicon types associated with the same carbapenemase or ESBL genes, such as *bla*_NDM_, *bla*_OXA-48_, and *bla*_CTX-M-15_. Given the dynamic nature of these processes and the severe outcomes associated with infections caused by these clones, accurate molecular typing of both the clones and genes is crucial. In conclusion, the molecular identification, through techniques such as Multi-Locus Sequence Typing (MLST) and Whole-Genome Sequencing (WGS), of specific high-risk clones is essential for implementing appropriate infection control measures and for preventing their epidemic dissemination.

## Data Availability

The original contributions presented in the study are publicly available. This data can be found at: https://www.ncbi.nlm.nih.gov/bioproject/PRJNA1193841/ (accessed on 20 January 2025).

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
