# Peer review of "Circulation of a Unique Klebsiella pneumoniae Clone, ST147 NDM-1/OXA-48, in Two Diverse Hospitals in Calabria (Italy)"

_antibiotics, 2025, doi:10.3390/antibiotics14020128_

Round 1

Reviewer 1 Report

Comments and Suggestions for Authors

Dear Authors,

I sincerely thank you for the opportunity to review this manuscript, that I read with great interest. In my opinion it is well structured but lacks syntactic precision, thus I have some suggestions in order to further improve its quality.

- Line 25: I suggest to write "Klebsiella pneumoniae" in its complete name the first time you mention it.

- Line 80: please extensively write the expression "New Delhi Metallo beta-lactamase" the first time you mention it. 

- Line 81: is there a particular reason why the Authors have chosen to present in December 2024 a work based on data collected one year ago? Do not they risk to be obsolete when published?

- Line 91: what does "forty nineteen" mean? I imagine the Authors meant "forty-nine", but please correct this expression since currently it is not a real number.

- Line 100: please write completely "Multi-Locus Sequence Typing" the first time you mention it. Same thing for "LPS O-antigen" in line 101.

- Line 166: please specify "in two hospital settings".

- Lines 237-239: Authors are pleased to precisely specify the inclusion and exclusion criteria adopted in this study (for example, the considered age range, which participants were included and which ones were excluded).

- Line 309: please specify the "species" mentioned in order to make this sentence more understandable.

In conclusion, I personally believe that this manuscript can be reconsidered after major revisions. I remain at your disposal.

Best Regards

Comments on the Quality of English Language

Dear Authors,

Here I report some suggestions in order to improve the English quality in your manuscript:

- Line 28: I suggest to replace "across hospitals in southern Italy" with "between two hospitals in southern Italy", in order not to make this sentence misleading.

- Line 58: I suggest to replace "each of ESKAPE pathogens" with "each one of the ESKAPE pathogens", since currently the syntax is not correct.

- Lines 58-61: I honestly cannot understand the meaning of this sentence. I strongly suggest to reformulate it, maybe with the help of a native speaker professional.

- Line 84: I suggest to start this sentence with "The aim of this work".

- Line 85: please add the verb "and to assess how both phenotypic and molecular characterization".

- Line 93: please replace "Among the antibiotics of interest tested" with "Among the tested antibiotics".

- Line 97: correct "as follow" with "as follows".

- Line 100: "appear" should be replaced with "appeared".

- Line 118: "all isolated" is not corrected. Please correct with "all the isolated..." and specify what you are talking about: all the isolated genes? Strains? Samples? Something else? Same correction in lines 119, 128, 138, 146, 168 and the following ones.

- Line 192: please correct "quinolone" with "quinolones".

I apologize for the huge number of these suggestions, but unfortunately the current English quality in this manuscript is not good at all. There are many expression that should be corrected and the syntax in general should be totally reviewed, thus I strongly suggest to the Authors to ask for the help of a native speaker professional.

Author Response

- Line 25: I suggest to write "Klebsiella pneumoniae" in its complete name the first time you mention it.

Done

- Line 80: please extensively write the expression "New Delhi Metallo beta-lactamase" the first time you mention it. 

Done

- Line 81: is there a particular reason why the Authors have chosen to present in December 2024 a work based on data collected one year ago? Do not they risk to be obsolete when published?

A: We appreciate the reviewer’s observation regarding the timing of data collection and the presentation of this work. The data presented in this manuscript originate from strains collected throughout 2023 from hospitals in the Calabria region. These strains were subsequently sent to Catania, Sicily, for further molecular analyses. The collaborative nature of this project, involving multiple institutions, required significant time for comprehensive molecular characterization, sequencing, data analysis, and manuscript preparation.

Given the complexity of these processes, the final manuscript was completed and submitted as soon as all aspects of the work were thoroughly addressed. Importantly, the study is intended as a retrospective epidemiological and molecular report, providing a detailed analysis of the strains collected over the specified timeframe. We believe this approach maintains the relevance and value of the findings, despite the retrospective nature of the study.

We hope this clarification addresses the reviewer’s concern.

- Line 91: what does "forty nineteen" mean? I imagine the Authors meant "forty-nine", but please correct this expression since currently it is not a real number.

A: The misunderstanding is due to a typo. The error has been corrected in the text and the intended number is forty-nine

- Line 100: please write completely "Multi-Locus Sequence Typing" the first time you mention it. Same thing for "LPS O-antigen" in line 101.

Done

- Line 166: please specify "in two hospital settings".

Done

- Lines 237-239: Authors are pleased to precisely specify the inclusion and exclusion criteria adopted in this study (for example, the considered age range, which participants were included and which ones were excluded).

A: The sentence has been changed to avoid misunderstanding. Criteria irrelevant to the purposes of the study, such as age and sex of the patients, were eliminated as inclusion criteria. The only inclusion criteria, for the subsequent in-depth molecular characterization, were: resistance to ceftazidime-avibactam and the co-presence of resistance genes such as NDM-1 and OXA-48, tested initially at hospitals using rapid tests. The entire section relating to strain collection and isolation, as well as the inclusion criteria, has been expanded, listing all the methods used by the various hospitals.

- Line 309: please specify the "species" mentioned in order to make this sentence more understandable.

A: The phrase was modified by adding “such as K. pneumoniae” To make the sentence more understandable.

Line 28: I suggest to replace "across hospitals in southern Italy" with "between two hospitals in southern Italy", in order not to make this sentence misleading.

Done

- Line 58: I suggest to replace "each of ESKAPE pathogens" with "each one of the ESKAPE pathogens", since currently the syntax is not correct.

Done

- Lines 58-61: I honestly cannot understand the meaning of this sentence. I strongly suggest to reformulate it, maybe with the help of a native speaker professional.

A: The entire manuscript was revised and corrected, from a syntactical, lexical and grammatical point of view, with the help of a native speaker, as suggested.

- Line 84: I suggest to start this sentence with "The aim of this work".

A: The sentence was modified as suggested

- Line 85: please add the verb "and to assess how both phenotypic and molecular characterization".

A: The sentence was modified as suggested

- Line 93: please replace "Among the antibiotics of interest tested" with "Among the tested antibiotics".

Done

- Line 97: correct "as follow" with "as follows".

Done

- Line 100: "appear" should be replaced with "appeared".

Done

- Line 118: "all isolated" is not corrected. Please correct with "all the isolated..." and specify what you are talking about: all the isolated genes? Strains? Samples? Something else? Same correction in lines 119, 128, 138, 146, 168 and the following ones.

All the senteces containing "all isolated" were modified as suggested by adding “The” and specifying the word “strains” for a better clarity.

- Line 192: please correct "quinolone" with "quinolones".

Done

The entire manuscript was revised and corrected, from a syntactical, lexical and grammatical point of view, with the help of a native speaker, as suggested.

Reviewer 2 Report

Comments and Suggestions for Authors

Manuscript title: A Klebsiella pneumoniae ST147 NDM-1/OXA-48 unique clone circulating in diverse hospitals in Calabria (Italy)

Manuscript ID/ File Name: antibiotics-3387050-peer-review-v1

Authors: Emanuele Nicitra et al.

Klebsiella pneumoniae, a ubiquitous pathogen, has evolved into a formidable multidrug-resistant strain (MDR-KP), which is increasingly responsibl for severe infections in healthcare settings. The emergence of resistance in Klebsiella pneumoniae is primarly driven by mechanisms such as the production of beta-lactamases and carbapenemases, the activity of efflux pumps, and the acquisition of resistance genes through horizontal gene transfer from other bacterial species. Patients in healthcare envirionments, particularly those with compromised immune systems or undergoing invasive medical procedures, are at heightened risk of infection. The transmission of MDR-KP is facilitated by inadequate infection control practices and the infections caused by MDR-KP present significant therpeutic challenges, often resulting in higher mortality rates, prolonged hospitalizations, and escalated healthcare costs.

In this context the study by Emanuele Nicitra et al. is relevant. The authors worked on a set of K. pneumoniae isolates which belonged to ST147 while harbouring NDM-1/OXA-48. They determined the AMS profile and went ahead with whole genome sequencing followed by bioinformatics analysis.

While the topic is important, there are several weaknesses which need to be addressed by the authors.

General:

1.      The authors stated that “A total of 49 OXA-48 and NDM-1 coproducing K. pneumoniae strains, resistant to ceftazidime-avibactam (CZA), were isolated from blood, urine, rectal swabs, bronchial aspirates, peritoneal fluid, and throat swabs” (Lines 234-235). If authors specifically isolated these organisms then it is not correct to claim that Klebsiella pneumoniae ST147 NDM-1/OXA-48 unique clone was circulating among hospitals. Had the authors isolated all K. pneumoniae and had they found the Klebsiella pneumoniae ST147 NDM-1/OXA-48 clone as predominant, then the authors’ assertion would have been justified. Therefore, please change the title.

2.      Another concern is that isolates’ genetic composition. How did the authors come to know during isolation that the organisms were coproducing NDM-1/OXA-48? Authors need to clarify this to have the manuscript in a coherent shape.

Materials and Methods:

1.      Authors did not establish the identity of the isolates. At present there are 27 species of Klebsiella (https://lpsn.dsmz.de/search?word=klebsiella). Unless clearly described, the scope of the manuscript remains difficult for readers.

2.      Authors please provide details of the term ‘all sequences’ in line 296.

Results and Discussion:

1.      Line(s) 91: The sentence is not clear.

2.      What were the criteria for designating a strain XDR?

3.      There is no mention of phylogenetic tree in the results section, though the authors showed one and described the construction in methods (Lines: 295-298)

4.      It is common convention for authors to provide access to genome sequence data for readers to appreciate the results better. However, the current bioproject (1193841) appears to be private and could not be accessed. This is a mandatory requirement for the data availability and authors need to work on this.

Conclusion:

1.      The used of the word ‘epidemic’ in line 305 is not derived from the study and therefore should be avoided.

2.      “the resistome of K. pneumoniae is subject to ... international travel to endemic regions” None of these were logically derived from the study reported. Therefore these should be avoided by the authors.

Author Response

General:

  1. The authors stated that “A total of 49 OXA-48 and NDM-1 coproducing K. pneumoniae strains, resistant to ceftazidime-avibactam (CZA), were isolated from blood, urine, rectal swabs, bronchial aspirates, peritoneal fluid, and throat swabs” (Lines 234-235). If authors specifically isolated these organisms then it is not correct to claim that Klebsiella pneumoniae ST147 NDM-1/OXA-48 unique clone was circulating among hospitals. Had the authors isolated all K. pneumoniae and had they found the Klebsiella pneumoniae ST147 NDM-1/OXA-48 clone as predominant, then the authors’ assertion would have been justified. Therefore, please change the title.

A: the isolation of the strains occurred through isolation from various anatomical sites following infection, as described, and from surveillance rectal swabs. All strains isolated at Calabrian hospitals were characterized for resistance to ceftazidime-avibactam and for the presence of resistance genes using a rapid method such as filmarray. Only the strains that presented the co-presence of resistance genes such as NDM-1 and OXA-48 as a characteristic in the first rapid tests carried out were sent to the University of Catania for further molecular investigations, such as whole genome sequencing which made it possible to confirm what was identified by the rapid tests, assign a specific sequence type and from this, through phylogenetic and bioinformatic investigations, trace back to a single circulating clone responsible for the small epidemic outbreak caused by ST-147 strains co-harboring NDM-1 and OXA-48 genes.

  1. Another concern is that isolates’ genetic composition. How did the authors come to know during isolation that the organisms were coproducing NDM-1/OXA-48? Authors need to clarify this to have the manuscript in a coherent shape.

 A: the sentence in question has been reworded to avoid misunderstandings and for better clarity. All strains isolated at Calabrian hospitals were characterized for resistance to ceftazidime-avibactam and for the presence of resistance genes using a rapid method such as filmarray. Only the strains that presented the co-presence of resistance genes such as NDM-1 and OXA-48 as a characteristic in the first rapid tests carried out were sent to the University of Catania for further molecular investigations, such as whole genome sequencing which made it possible to confirm what was identified by the rapid tests, assign a specific sequence type and from this, through phylogenetic and bioinformatic investigations, trace back to a single circulating clone responsible for the small epidemic outbreak caused by ST-147 strains co-harboring NDM-1 and OXA-48 genes.

Materials and Methods:

  1. Authors did not establish the identity of the isolates. At present there are 27 species of Klebsiella (https://lpsn.dsmz.de/search?word=klebsiella). Unless clearly described, the scope of the manuscript remains difficult for readers.

A: We thank the reviewer for asking us this question that allowed us to better clarify some aspects of the manuscript.

We have inserted in both the methods and results some sentences that better clarify the work carried out.

The isolates were initially identified at the hospitals of Cosenza and Catanzaro, subsequently only  those resulting K. pneumoniae OXA48 and NDM 1 were sent to our laboratory for further investigations both phenotypic and molecular.

  1. Authors please provide details of the term ‘all sequences’ in line 296.

A: we better explain in the text using this sentence:"Phylogenetic tree was constructed using mafft (v7.525)[18,19] with 100 iterations, followed by fasttree(v.2.1.11)[20]. The illustration of the tree has been created with the R packages: ape, ggtree, RColorBrewer, ggplot2 and ggnnewscale [30][31]."

Results and Discussion:

  1. Line(s) 91: The sentence is not clear.

A: The sentence was modified as follows for better clarity: “Forty-nine clinical samples of K. pneumoniae were included in the study: 48/49 (97,9%) were CZA resistant K. pneumoniae (CZA-R Kp) strains and 1/49 (2,04%) was a CZA susceptible K. pneumoniae (CZA-S Kp) strain (CZ17).”

  1. What were the criteria for designating a strain XDR?

A: we used the definition reported by S. Navon-Venezia, K. Kondratyeva, e A. Carattoli, «Klebsiella pneumoniae: a major worldwide source and shuttle for antibiotic resistance», FEMS Microbiol. Rev., vol. 41, fasc. 3, pp. 252–275, mag. 2017, doi: 10.1093/femsre/fux013.

  1. There is no mention of phylogenetic tree in the results section, though the authors showed one and described the construction in methods (Lines: 295-298)

A: a paragraph called “phylogenetic tree” was added to the “results” section.

  1. It is common convention for authors to provide access to genome sequence data for readers to appreciate the results better. However, the current bioproject (1193841) appears to be private and could not be accessed. This is a mandatory requirement for the data availability and authors need to work on this.

A: Here we provide the link of the project: https://dataview.ncbi.nlm.nih.gov/object/PRJNA1193841?reviewer=s8k8d7jfi9a8j90h2ipbui42vp

Conclusion:

  1. The used of the word ‘epidemic’ in line 305 is not derived from the study and therefore should be avoided.

A: The word epidemic was deleted in the line.

  1. “the resistome of K. pneumoniae is subject to ... international travel to endemic regions” None of these were logically derived from the study reported. Therefore these should be avoided by the authors.

A:  thanks to the reviewer we have modified this sentence

The entire manuscript was revised and corrected, from a syntactical, lexical and grammatical point of view, with the help of a native speaker.

Reviewer 3 Report

Comments and Suggestions for Authors

Dears authors,

Thank you for the opportunity to review your manuscript. I found your work very important, interesting, and well done. My comments relate to some presentation of your findings, which might be improved by the addition of table and formal statistical tests to support some of the conclusions reached in your discussion. Similarly, introduction of the specimens clinical/demographic characteristics would improve the reader's orientation to your findings. In addition to these comments you will fine some minor points below:

Line 52: Citation for ESKAPE

Line 55: I think "contrast" should be "combat"

Line 75: Need a citation for the resistance genes.

Line 82: "2 hospitals in southern Italy, Calabria" I would just say, "from two hospitals in Calabria, Italy"

Line 93: "Among the antibiotics of interest tested there are also meropenem (MEM)" is somewhat non-idiomatic. Perhaps "We tested the following antibiotics of interest: merospenem (MEM), ..."

Line 167: You mention rectal swab as a specimen source with higher rates, but you haven't done any statistics on it. It may be worth summarizing some of these demographics/ clinical characteristics in a table, or introducing them in the results to discuss at length in the discussion. I found myself looking at Figure 1 and trying to understand some of these key features of the specimens themselves and couldn't (i.e., I was counting the colored boxes). It might make it easier for the reader to place some of these data in a table and then conduct some of the other inferential statistics of interest and have them somewhere in a supplement (i.e., a chi-square for the presence of a resistance marker by specimen location).

Line 175: This risk statement is not justified without formal statistical testing.

185: just a stylistic point, I would introduce the two hospital names earlier.

Discussion: More generally, I recommend breaking up your discussion into multiple paragraphs. Each paragraph should echo a paragraph in the results, summarizing the main finding and putting them into context with the existing literature. As written, I am having to pull out your findings from what others have done. For example, you could break a paragraph at line 182 for your discussion of the restiome, I would also break out your key recommendations in the final lines into its own paragraph and expand a little more (i.e., what should be we be thinking about here).

Line 237: :ncluded in the study based on criteria such as age, sex, tests per- formed at the time of hospitalization' If age/sex were part of the study criteria and cohort selection, you need to detail what those criteria were. For example, specimens from adults only? 

341: Worth mentioning that the WGS and should be rapid/point of care for patient care vs just infection control.

A note on the supplements: I was only able to see Table S1 and no other materials.

Author Response

Line 52: Citation for ESKAPE

A: A citation was added as suggested.

Line 55: I think "contrast" should be "combat"

A: done

Line 75: Need a citation for the resistance genes.

A: Done

Line 82: "2 hospitals in southern Italy, Calabria" I would just say, "from two hospitals in Calabria, Italy"

A: The sentence was modified as suggested

Line 93: "Among the antibiotics of interest tested there are also meropenem (MEM)" is somewhat non-idiomatic. Perhaps "We tested the following antibiotics of interest: merospenem (MEM), ..."

A: The sentence was modified as suggested

Line 167: You mention rectal swab as a specimen source with higher rates, but you haven't done any statistics on it. It may be worth summarizing some of these demographics/ clinical characteristics in a table, or introducing them in the results to discuss at length in the discussion. I found myself looking at Figure 1 and trying to understand some of these key features of the specimens themselves and couldn't (i.e., I was counting the colored boxes). It might make it easier for the reader to place some of these data in a table and then conduct some of the other inferential statistics of interest and have them somewhere in a supplement (i.e., a chi-square for the presence of a resistance marker by specimen location).

A: We thank the reviewer for this suggestion but we have no further data regarding the patients and we cannot do any statistical analysis. We regret not being able to resolve this issue.

Line 175: This risk statement is not justified without formal statistical testing.

A: the sentence in question is related to the study cited and useful for the purposes of the discussion and not to the one being reviewed. The sentence has been changed and a citation was added at the end of the sentence to avoid misunderstandings [line 183].

185: just a stylistic point, I would introduce the two hospital names earlier.

A: The sentence was modified as suggested [lines 190-193].

Discussion: More generally, I recommend breaking up your discussion into multiple paragraphs. Each paragraph should echo a paragraph in the results, summarizing the main finding and putting them into context with the existing literature. As written, I am having to pull out your findings from what others have done. For example, you could break a paragraph at line 182 for your discussion of the restiome, I would also break out your key recommendations in the final lines into its own paragraph and expand a little more (i.e., what should be we be thinking about here).

A: The discussion section has been split into subsections as suggested by the reviewer

Line 237: included in the study based on criteria such as age, sex, tests per- formed at the time of hospitalization' If age/sex were part of the study criteria and cohort selection, you need to detail what those criteria were. For example, specimens from adults only? 

A: The sentence has been changed to avoid misunderstanding. Criteria irrelevant to the purposes of the study, such as age and sex of the patients, were eliminated as inclusion criteria. The only inclusion criteria, for the subsequent in-depth molecular characterization, were: resistance to ceftazidime-avibactam and the co-presence of resistance genes such as NDM-1 and OXA-48, tested initially at hospitals using rapid tests.The entire section relating to strain collection and isolation, as well as the inclusion criteria, has been expanded, listing all the methods used by the various hospitals.

341: Worth mentioning that the WGS and should be rapid/point of care for patient care vs just infection control.

A: Due to issues of timing and high costs, as well as the need for highly qualified staff for whole genome sequencing and bioinformatics analysis, we do not believe it is appropriate to mention WGS as a rapid point of care, but as a useful tool, at the moment , for the control of infections and for the tracing of resistance and virulence determinants, in this specific case.

A note on the supplements: I was only able to see Table S1 and no other materials.

A: table S1 is the only material showed on the supplements.

The entire manuscript was revised and corrected, from a syntactical, lexical and grammatical point of view, with the help of a native speaker.

Round 2

Reviewer 1 Report

Comments and Suggestions for Authors

Dear Authors,

I sincerely thank you for the opportunity to review the revised version of your manuscript. I also thank you for your precise and punctual answers to all my doubts.

In my opinion, you manuscript is now ready for publication. I would just suggest you to edit the title like this: "A Klebsiella pneumoniae ST147 NDM1/OXA 48 unique clone circulating in two diverse hospitals in Calabria (Italy)", in order to clarify the number of the involved hospitals.

I remain at your disposal.

Best Regards

ly)

Author Response

Dear Authors,

I sincerely thank you for the opportunity to review the revised version of your manuscript. I also thank you for your precise and punctual answers to all my doubts.

In my opinion, you manuscript is now ready for publication. I would just suggest you to edit the title like this: "A Klebsiella pneumoniae ST147 NDM1/OXA 48 unique clone circulating in two diverse hospitals in Calabria (Italy)", in order to clarify the number of the involved hospitals.

A: Dear reviewer, thank you for your suggestion. The authors agreed with you and they changed the title as suggested.

Reviewer 2 Report

Comments and Suggestions for Authors

The authors have attempted to improve the manuscript considerably and the manuscript is in better shape now. However, there are three issues which remained unaddressed.

1. The Bioproject link provided by the authors is still private. And this must be made public for the sake of the readers.

2. In Conclusion section the desired amendments were not addressed. Please address this

  1. “the resistome of K. pneumoniae is subject to ... international travel to endemic regions” None of these were logically derived from the study reported. Therefore these should be avoided by the authors.

A: thanks to the reviewer we have modified this sentence"

3. The manuscript does require English language check once more.

Please see the new paragraph on phylogenetic analysis Lines 165 - 170. The tenses in this paragraph are mixed up.

Comments on the Quality of English Language

3. The manuscript does require English language check once more.

Please see the new paragraph on phylogenetic analysis Lines 165 - 170. The tenses in this paragraph are mixed up.

Author Response

The authors have attempted to improve the manuscript considerably and the manuscript is in better shape now. However, there are three issues which remained unaddressed.

  1. The Bioproject link provided by the authors is still private. And this must be made public for the sake of the readers.

A: the link was update: https://www.ncbi.nlm.nih.gov/bioproject/PRJNA1193841/

  1. In Conclusion section the desired amendments were not addressed. Please address this

    The authors have attempted to improve the manuscript considerably and the manuscript is in better shape now. However, there are three issues which remained unaddressed.

    1. The Bioproject link provided by the authors is still private. And this must be made public for the sake of the readers.

    A: the link was update: https://www.ncbi.nlm.nih.gov/bioproject/PRJNA1193841/

    1. In Conclusion section the desired amendments were not addressed. Please address this  A: the review can found the bioproject open. 
      1. “the resistome of K. pneumoniae is subject to ... international travel to endemic regions” None of these were logically derived from the study reported. Therefore these should be avoided by the authors.

      A: Dear reviewer, the authors modified the sentence to avoid any statement non logically derived from the study reported, asserting only what is widely demonstrated and reported in the literature.

      1. The manuscript does require English language check once more.

      Please see the new paragraph on phylogenetic analysis Lines 165 - 170. The tenses in this paragraph are mixed up.

      A: Dear reviewer, thank you for your suggestion. The new paragraph has been revised from a stylistic and linguistic point of view once more.

Reviewer 3 Report

Comments and Suggestions for Authors

My congratulations to the authors. The revised manuscript is excellent. 

Author Response

My congratulations to the authors. The revised manuscript is excellent. 

A: Dear reviewer, the authors thank you very much for your time and suggestions.